# Calcifying Tendinopathy of the Rotator Cuff: Barbotage vs. Shock Waves: Controlled Clinical Trial Protocol (BOTCH)

**DOI:** 10.3390/healthcare13010014

**Published:** 2024-12-24

**Authors:** Javier Muñoz-Paz, Fiorella Liz Piaggio-Muente, Sebastián Acosta-Salvador, Diego A. Gómez-Flores, Ana Belén Jiménez-Jiménez, María Nieves Muñoz-Alcaraz, Fernando Jesús Mayordomo-Riera

**Affiliations:** 1Interlevel Clinical Management Unit of Physical Medicine and Rehabilitation, Reina Sofía University Hospital, Córdoba and Guadalquivir Health District, 14011 Córdoba, Spain; javier.munoz.paz.sspa@juntadeandalucia.es (J.M.-P.); piaggio.fiorella.sspa@juntadeandalucia.es (F.L.P.-M.); sebastian.acosta.sspa@juntadeandalucia.es (S.A.-S.); diegoa.gomez.sspa@juntadeandalucia.es (D.A.G.-F.); anab.jimenez.jimenez.sspa@juntadeandalucia.es (A.B.J.-J.); fernandoj.mayordomo.sspa@juntadeandalucia.es (F.J.M.-R.); 2Maimónides Biomedical Research Institute of Cordoba (IMIBIC), Reina Sofia University Hospital, University of Córdoba, 14004 Córdoba, Spain; 3Department of de Applied Physics, Radiology, and Physical Medicine, University of Córdoba, 14004 Córdoba, Spain

**Keywords:** ultrasound-guided percutaneous irrigation, extracorporeal shockwave therapy, calcific tendinopathy, rotator cuff, shoulder pain

## Abstract

**Background**: Shoulder pain is a very common health issue among adults, being 8% due to calcifying tendinopathies (CT) of the shoulder. The evolutionary process of this lesion can be classified according to Bianchi Martinoli, depending on the ultrasound appearance. In 50% of cases, with first-line treatments, they resolve spontaneously. However, in the remaining 50%, they become chronic, requiring other lines of treatment, such as shock waves (ESWT) or ultrasound-guided barbotage (US-PICT). **Objectives**: The objective focuses on comparing the improvement in pain using the visual analgesic scale (VAS) and shoulder joint balance (ROM) in patients with CT based on the treatment received, stratifying according to the characteristics of the injury, with the aim of protocolizing said treatment. **Methods**: Randomized analytical controlled clinical trial in blocks with two arms according to the Bianchi Martinolli classification (I or II/III) in 56 patients affected by chronic pain by CT. The decision to treat will be made randomly 1:1, based on the treatment assigned to the previous patient. Results will be evaluated in three moments (1, 3, and 6 months). The following variables will be collected: VAS, Lattinen test, ROM (flexion, abduction, external and internal rotation), patient global improvement impression scale (PGI-I), global improvement impression scale (CGI-C). **Discussion**: The use of ESWT or US-PICT as treatments is a widely used practice in the daily life of this pathology. However, despite knowing that both treatments are useful in chronic CT, there are no known data or protocols by which one therapy is chosen over another, much less the influence that the evolutionary stage can have of the injury in the results obtained.

## 1. Introduction

### 1.1. Background and Rationale

Shoulder pain is a very frequent complaint among adults. A large number of these are due to calcifying tendinitis (CT) of the rotator cuff, with prevalences between 3 and 8% in the general population, especially in women aged 30–60 years [1].

Among risk factors that we can assume, many consider that “hormonal abnormalities and/or autoimmune diseases such as hypothyroidism, diabetes mellitus and rheumatoid arthritis” [2] play an important role in the development and chronicity of CT [3]. The natural course of CT pathogenesis is currently incomplete. Cook et al. published the continuum model of changes that a tendon undergoes when it is damaged [4]. The first hypothesis is that the calcium hydroxyapatite deposition is the pathophysiological basis of CT, causing inflammation, pain, and functional impotence. However, this deposit, as opposed to the first theories that described its appearance due to tendon degeneration, is not currently supported [5].

In this pathology, what is known is the evolutionary course passing through three phases according to Chianca et al. [6]:1. Pre-calcified → 2. Calcified (edema and pain) →3. Post-calcification

For diagnosis, physical examination and imaging-based tests are strictly enforced, and, among them, radiography and ultrasound of the rotator cuff are the gold standard tests. Having used these tests, two classifications have been created that relate to the evolutionary course described above.

Radiological classification by Gärtner and Heyer [7].

Circumscribed, dense, in formation.Dense/translucent, circumscribed/cottony outlineCloud-like and translucent, in resorptive phase.

Ultrasound classification by Bianchi Martinolli (Figure 1) [8].

The symptomatology of this disease has a wide range of variability among patients. Speaking of general numbers, 50% of the patients recover spontaneously or following conservative guidelines in the first 3 months and 20% during the first year. The remaining 30% develop repeated pain for months in periods exceeding several years [9].

Having explained the most relevant data on which CT is based, we focus on treatment. The initial therapy, being a self-limited disease, is focused on symptomatology control with anti-inflammatory drugs and oral analgesics, being able to resort to glucocorticoid injections in the rotator cuff in those cases in which inflammation is observed.

This protocol, whose acronym comes from the Spanish words, “barbotage, shock waves, tendinopathy, calcium and shoulder” (BOTCH), will focus its research on the comparison of second-line treatments in CT, such as ultrasound-guided barbotage (US-PICT) and shock waves (ESWT).

This protocol has followed the established quality standards marked by the “SPIRIT 2013 Declaration” [10]. The group of individuals that make up the BOTCH study team declare that they have no conflicts of interest.

### 1.2. Starting Hypothesis

Patients with CT, in its different degrees, who have received US-PICT as treatment obtain the same visual analgesic scale (VAS) and joint balance (ROM) differences at 1 month, 3 months, and 6 months as patients who received ESWT as treatment in the population.

## 2. Objectives

### 2.1. Main Objective

-The main objective is to compare the mean differences in VAS at 1 month, 3 months, and 6 months according to the treatment received, US-PICT or ESWT, in patients with CT, according to the evolutionary stage of the lesion based on the Bianchi Martinolli classification.

### 2.2. Secondary Objectives

-Knowing if there are variations in functionality with the Latineen test, ROM, patient global improvement impression scale (PGI-I), and global improvement impression scale (CGI-C).

## 3. Trial Design

The trial protocol is registered in clinical trials with such number NCT06528756 (https://clinicaltrials.gov/study/NCT06528756, accessed on 1 August 2024).

### 3.1. Study Setting

The protocol carried out was as follows:

First Phase: In this first phase, patients will be collected according to the criteria set out above for a period of 12 months. During this time, the patients will be assigned one of the two possible treatments to be received, and after signing the appropriate informed-consent form, they will be divided into groups according to treatment (ESWT vs. US-PICT), stratified according to the Bianchi Martinolli classification: (I/II or III). For this study, an ultrasound machine with a 15–6 MHz linear probe was used.

The decision for each treatment will be made randomly, dividing the patients in a 1:1 ratio according to whether they come to the initial consultation and the previous patient has received one or the other treatment.

Second Phase: Subsequently, three revisions will be carried out. One at 1 month, 3 months, and another at 6 months after the end of therapy. In these, the variables will be taken again as indicated in the table of variables.

Both phases, as well as the variables that are collected, are explained in Figure 1 and in Table 1, presented below.

### 3.2. Eligibility Criteria

Inclusion criteria:
-Patients aged 30–60 years.-Shoulder pain lasting more than 3 months.-Radiographic and ultrasound visualization in both planes of calcification.-Sizes > 5 mm (mm) [9].
Exclusion criteria:
-Patients not meeting inclusion criteria.-Presence of another evident cause of pain (joint degeneration, capsulitis, rotator cuff tendon ruptures…).-Patient who has previously received ESWT or US-PICT.-Contraindication of therapies: infection, drug allergies, cancer….


### 3.3. Who Will Take Informed Consent?

The patient inclusion and selection process will be carried out by Javier Muñoz-Paz (J.M.-P.).

He will explain the most important points of the procedure to be followed, and subsequently, if the patient wishes, the appropriate informed consent would be signed.

### 3.4. Interventions

Once the patient is included in the protocol, they will be assigned a treatment to follow, as indicated in the study-setting section.

#### 3.4.1. ESWT

This will be administered using a 15 mm transmitter in continuous mode with pressure at 3 bar, frequency at 12 Hz, and 3000 impacts. The number of sessions will be between 4 and 8, according to the patient’s clinical condition, with a rest between sessions of 5 to 10 days.

#### 3.4.2. US-PICT

This therapy will be performed with one session in which the following actions will be performed:-Firstly, a suprascapular nerve block will be performed with corticosteroid (Triamcinolone acetonide 40 mg/mL suspension for injection) and anesthetic (3 mL of bupivacaine) in an ultrasound-guided manner.-Secondly, the calcification will be infiltrated with 5 mL of local lidocaine and preloaded physiological saline solution under ultrasound guidance.-Finally, repeated aspirations with physiological saline solution will be performed to extract calcium under ultrasound guidance, and 0.5 mL of corticosteroid will be left in the bag at the end.

### 3.5. Withdrawals or Dropouts from the Trial

Possible discontinuities in visits, abandonments, losses of patients, as well as their reasons will be recorded in an incident record. If the patient shows their intention to abandon the trial for any reason, they will be withdrawn from it.

Those patients who, due to the clinical criteria of the principal investigator, have to abandon said trial may also be withdrawn, without this entailing any aggravation regarding the patient’s treatment.

The SPSS database will be created only with patients in whom all the indicated variables are collected, not including patients with missing data.

### 3.6. Sample Size [11]

The GRANMO online calculator was used to calculate the sample size. An alpha risk of 0.05 and a statistical power greater than 0.8 was accepted in a bilateral contrast, requiring 14 subjects in group 1 and 14 in group 2 to detect a difference equal to or greater than 2 units of the VAS. The common standard deviation was assumed to be 1.7 [11]. A loss to follow-up rate of 20% was estimated. As the study presents stratification in two arms, the total number of patients needed was 56.

### 3.7. Recruitment

Patients from both groups will be recruited at the Physical Medicine and Rehabilitation Unit of the Reina Sofía University Hospital (HURS) (Córdoba, Spain) by Dr. J.M.-P.

Referral from the initial consultation to the indicated treatment will be made by the means used in routine clinical practice, such as DIRAYA.

The ESWT technique will be performed by Fst. D.A.G.-F. (Diego A. Gómez Flores). The US-PICT technique will be performed jointly by Dr. J.M.-P., Dr. F.L.P.-M. (Fiorella Liz Piaggio Muente), and Dr. S.A.-S. (Sebastián Acosta Salvador).

Once the treatment is finished, the team members will notify Dr. J.M.-P. through internal mechanisms so that control of subsequent reviews is carried out at the indicated times. These reviews will be carried out jointly by specialist doctors in physical medicine and rehabilitation, Dr. J.M.-P., Dr. F.L.P.-M., and Dr. S.A.-S.

### 3.8. Assignment of Interventions

#### 3.8.1. Allocation

The process of assigning each treatment will be carried out with the aim of achieving the same number of patients in each group.

Therefore, patients, after a first consultation, will be assigned a therapy by block randomization in a 1:1 ratio between ESWT and US-PICT.

The first patient presented with both options of treatments will be the one who determines which therapy is applied first.

Randomization is established by randomization blocks to avoid numerical inequality between the different groups and to obtain the sample size needed in each group.

#### 3.8.2. Blinding

This protocol does not include blinding due, mainly, to the large technical differences that exist between the therapies, which makes blinding difficult for both the personal physician and the patient.

In order for this protocol to be unbiased, we will create a database in SPSS. 21, and the following statistical study will be carried out by A.B.J.-J. (Ana Belén Jiménez Jiménez), who will not have contact with the patients and will not know what type of intervention was assigned to each patient.

These data will be previously blinded by J.M.-P., so that blinding is applied to the study in its final phase.

### 3.9. Adverse Event

All adverse events will be recorded in the DIRAYA system. They will then be uploaded to the SPSS database for critical reporting.

The appearance of an adverse effect, which implies abandonment of the study, will be reported directly to J.M.-P. He will proceed to activate the internal mechanisms that the HURS has through the HURS Adverse Events Notification Platform.

### 3.10. Limitations

This trial may be affected by certain limitations, including the following:-Poor adherence to the proposed interventions, as well as to patient follow-up.-Impossibility of blinding both the medical staff/physiotherapist and the patient, due to the technical difference between treatments.-Care overload that slows down the time from diagnosis to treatment, which in-creases the time it takes to collect patients.

## 4. Data Collection and Management

The principal investigator of the work may contact the rest of the team to resolve missing data and avoid the loss of patients.

The information generated will be collected from the clinical reports created in the DIRAYA system, by the main research. Dr. J.M.-P., Dr. F.L.P.-M., Dr. S.A.-S., and Fst. D.A.G.-F. will have access to the information stored in DIRAYA.

The elimination of personal and demographic information will be carried out, as well as the assignment of an alphanumeric code to each patient.

All this information will be dumped into an Excel database to later be transferred to the SPSS program. It will be protected and stored in a place with adequate security. Only the main investigator, management and the Clinical Research and Ethics Committee that approved the trial will have access to the collected data set.

All this personal information will be kept by the main researcher during the entire period of the study and for a maximum of 5 years, as established by Spanish national legislation.

## 5. Statistical Methods

The information collected will be synthesized and loaded into the SPSS.21 program to obtain the results. For these, the following actions will be carried out:
For an independent data design:
-Two groups: Student’s T-test or Mann–Whitney U-test.-More than two groups: Analysis of variance or Kruskal–Wallis “H” test.For a paired data design:
-Two groups: Student’s t-test for paired data or Wilcoxon test.-More than two groups: Analysis of variance for repeated measures or Friedman’s test.

Main variables: VAS, Test de Latineen y variables, PGI y CGI.

Grouper variable: ESWT vs. US-PICT

To correlate two quantitative variables with each other, the Pearson Linear Correlation Coefficient (r) will be used. At all times, for multiple comparisons, a test (Bonferroni, Finner…) will be used to correct the “ *p*” value. To perform the multiple analysis, a multiple linear regression analysis will be used.

## 6. Results

VAS [7,12,13]

The VAS “is a validated, subjective measure for acute and chronic pain”. This is a scale that allows for measuring pain intensity with maximum reproducibility.

It consists of a horizontal line of 10 cm, at the ends of which are the extreme expressions of a symptom. On the left is the absence or lower intensity, and on the right is the greatest intensity. The patient is asked to mark on the line the point that indicates the intensity. The minimally detectable differences for the symptom status to be acceptable is 2 to 3 points.

The VAS will be measured in the initial consultation and at 1, 3, and 6 months after the intervention. A new variable VAS difference (VASDIF) will be created later, with the data obtained, according to the following formula in order to evaluate the variation with the initial VAS levels (Figure 2).
VAS_in_ − VAS_xm_ = VASDIF_xm_

                               [Time Frame: 6 months]

Latineen test [12,14]

The Latineen test is a widely used tool for pain assessment. It was validated in Spain as a tool to measure the degree of affectation of the patient with chronic pain. It is useful to score five different items within this scale that score from 0 to 4. This scale was transferred to 1, 3, and 6 months:Pain intensity.Frequency of pain.Consumption of painkillers.Degree of disability.Hours of sleep.

                               [Time Frame: 6 months]

Correlation of VAS levels and Latineen test:VAS of 0–2.5 → IL = 6 VAS of 2.5–5 → IL = 10 VAS of 5–7.5 → IL 13 VAS of 7.5–10 → IL = 14

ROM [15,16]

The evaluation of range of motion is a basic practice in the study of shoulder pathologies. ROM should always be measured actively and passively to discard different entities such as adhesive capsulitis.

In our case, shoulder movements were evaluated actively and passively to determine flexion, abduction, external rotation with the arm at 90° of abduction, and internal rotation with the arm at 90° of abduction measured with PLURIMETER inclinometer. The measurement was conducted with the patient actively sitting, and if limitation was found, the patient was positioned in a supine position to eliminate the gravity factor.

                               [Time Frame: 6 months]

PGI-I [17,18]

PGI scale was published in 1976 by the National Institute of Mental Health (USA). In its PGI-I section, it mainly measures the change or improvement of the disease after an intervention.

The PGI-I consists of a single question that asks the patient to classify the relief obtained with the treatment according to a 7-point Likert scale:

1 = Significantly improved.

2 = Much improved.

3 = Minimally improved

4 = No change.

5 = Minimally worsen.

6 = Much worsen.

7 = Significantly worsen.

                               [Time Frame: 6 months]

CGI-C [18]

CGI-C: the physician compares the patient’s initial clinical condition with the current condition: Compared to the patient’s condition at the time of admission to the project [prior to the start of medication], this patient’s condition is as follows:

1 = Very much better since the start of treatment.

2 = Much improved.

3 = Minimally improved

4 = No change since baseline (the start of treatment).

5 = Minimally worse.

6 = Much worse.

7 = Very much worse since the start of treatment.

                               [Time Frame: 6 months]

Currently, the PGI -I and CGI-C scales do not have validation for tendon pathologies of the rotator cuff. However, we believe that they can be very useful to us.

We consider that it is important to take into account the results self-reported by the patients themselves in this type of pathologies.

In Spain, some consensus documents have been prepared on the approach to chronic non-cancer pain in which they recommend the use of these scales [18]. In addition, systematic reviews and meta-analyses can be found, such as that by Varag Abed et al., in which they report that “Multiple patient-reported outcome measures have been used to evaluate shoulder function, but it is unknown what they are. the most effective” [19].

## 7. Dissemination Plans

The study results will be communicated to clinical stakeholders and discussed in communications at scientific events and through publications in peer-reviewed scientific journals. The final results will also be communicated to the health professionals involved in the care of this type of patient through the internal communication channels of the Andalusian Health Service.

## 8. Discussion

For many years, the use of ESWT has been a controversial practice in relation to its use in CT due to the lack of studies. The mechanism of action of this technique focuses on the “analgesic, osteogenic, neovascular and tissue repair” [20] effect that microwaves produce in the surrounding tissue. After the study published by Gerdesmeyer et al., which showed that, even with low-energy ESWT, it was possible to dissolve these calcifications [21], its use and new publications have been encouraged. In 2016, the Spanish Society of Shock Wave Treatments (SETOC) managed to include CT as an approved standard indication for the use of ESWT [22].

For many authors, this treatment “has proved effective in functional improvement and pain reduction of calcific tendinitis” [7]. However, a recent Cochrane review on the use of this technique in rotator cuff pathologies, including those caused by calcium deposits, concluded that “the evidence currently available” is “low to moderate certainty”, and that there were very few clinically important benefits” [23].

Regarding the use of US-PICT, this technique is increasingly in use. Being a minimally invasive technique whose foundation is centered on the washing and aspiration of the calcium material that damages the tendon [24], there are multiple studies that support this practice, recognizing it as “an effective and safe treatment in CT, since it produces a significant clinical and radiological improvement” [9]. A study carried out in 2022, which sought to retrospectively study the results of this technique, culminated in demonstrating a significant improvement in pain in patients treated with this technique [25]. However, it is true that the long-term efficacy is up for debate [26].

Several studies have been conducted to determine which type of therapy is most effective in each case, with very controversial conclusions. In the 2014 systemic review by Louwerens et al., it was shown that both the use of high-energy ESWT and US-PICT were effective treatments in chronic cases of CT for patients with symptomatology longer than 6 months; however, it gave greater effectiveness to ESWT [27]. However, in 2014, Yang-Soo Kim, MD et al. published a randomized study which concluded that “ultrasound-guided needling treatment was more effective in restoring function and short-term pain relief” [28].

In patients with CT refractory to first-line treatments, the use of ESWT or US-PICT plays an important role. Studies, such as that one of Angileri et al., corroborate “the reasonable need to test US-PICT and ESWT as treatment” prior to evaluating a surgical option [29].

The main problem with this type of therapy is the lack of studies that support its use in a protocolized manner. In addition, it is impossible to find comparisons between these techniques based on the stage of the injury. Perhaps the clinical diversity and the great differences between the mechanisms of action of each service are the cause.

## 9. Conclusions

CT is a common entity in the general population and in the clinical practice of the specialty of physical medicine and rehabilitation. The appearance of new treatment therapies prior to a surgical intervention in patients with refractory CT opens a wide field of therapeutic possibilities.

Although the use of ESWT and US-PICT is increasingly widespread in daily clinical practice, more quality studies are needed to create a unique action mechanism, in which the specific use of each technique is taken into account. The evolutionary state of the lesion is a possible influencing factor, in order to propose such protocolization.

## Data Availability

The database obtained for the analysis of this study will be made available to those researchers who request it. When the data are analyzed, they will be published in full detail in open access, except for those that affect the confidentiality of the participants. On this database, any type of personal information that allows patients to be identified will be eliminated.

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
