# Peer review of "Calcifying Tendinopathy of the Rotator Cuff: Barbotage vs. Shock Waves: Controlled Clinical Trial Protocol (BOTCH)"

_healthcare, 2024, doi:10.3390/healthcare13010014_

Round 1

Reviewer 1 Report

Comments and Suggestions for Authors

This study addresses an important clinical issue, calcifying tendinopathy (CT) of the shoulder, by comparing two widely used interventions. The study design is clear, and the effort to standardize treatment protocols is commendable.

Few comments:

1.      The authors state that blinding is not possible due to technical differences between the interventions. This is understandable, but it is a limitation. The authors should elaborate on how they plan to minimize bias that might arise.

2.      Line 160: The common standard deviation was assumed to be 1.7. Provide a source or previous study that supports this assumption.

3.      The use of the Patient Global Improvement Impression Scale (PGI-I) and the Clinical Global Impression Scale (CGI-C) is not validated for tendinopathy. The authors acknowledge this but does not justify their inclusion or how this affects data interpretation. Please provide more information here.

Author Response

Dear reviewer:
Thank you very much for taking the time to review this manuscript. We also thank you very much for so kindly acknowledging the efforts we have made to standardize our intervention.
Please find the detailed responses below and the corresponding corrections highlighted in red in track changes in the re-submitted files.

Comments 1: The authors state that blinding is not possible due to technical differences between the interventions. This is understandable, but it is a limitation. The authors should elaborate on how they plan to minimize bias that might arise.
Response 1: Thank you very much for referring to this aspect. We completely agree with you, that is why we have added a limitations section (3.10) in our study between lines 227 – 234. The possibility of bias will also be attempted to be minimized through blinding of the data analysts and the statistician. This blinding has been highlighted in Scheme 1 and explained in greater detail in the blinding point (3.8.2) between lines 211 – 219.

Comments 2: Line 160: The common standard deviation was assumed to be 1.7. Provide a source or previous study that supports this assumption.
Response 2: We appreciate that you pointed out to us the importance of this information. Said reference has been added on line 170, which corresponds to:
Louwerens JKG, Sierevelt IN, Kramer ET, Boonstra R, Van den Bekerom MPJ, Van Royen BJ, et al. Comparing Ultra-sound-Guided Needling Combined With a Subacromial Corticosteroid Injection Versus High-Energy Extracorporeal Shockwave Therapy for Calcific Tendinitis of the Rotator Cuff: A Randomized Controlled Trial. Arthroscopy. 2020; 36(7):1823-33

Comments 3: The use of the Patient Global Improvement Impression Scale (PGI-I) and the Clinical Global Impression Scale (CGI-C) is not validated for tendinopathy. The authors acknowledge this but does not justify their inclusion or how this affects data interpretation. Please provide more information here.
Response 2: We appreciate that you pointed out this aspect, which can undoubtedly represent a limitation of the design in our research and as such we have added between lines 357 - 365 a more detailed explanation of why we consider it appropriate to use these scales. Currently, the PGI-I and CGI-C scales do not have validation for rotator cuff tendon pathologies. However, we believe that they can be very useful to us.
     We consider that it is important to consider the results self-reported by the patients themselves in this type of pathology. 
      In Spain, some consensus documents have been prepared on the approach to chronic non-cancer pain in which they recommend the use of these scales. (17) Additionally, systematic reviews and meta-analyses can be found such as that by Varag Abed et al., in which they report that “Multiple patient-reported outcome measures have been used to evaluate shoulder function, but it is unknown which ones.” are the most effective.” (18)

17. Sánchez J, Tejedor A, Carrascal R. In Atención al Paciente con Dolor Crónico No Oncológico en AP. Sociedad Española de Médicos Generales y de Familia en AP. Spain, 2023; pp. 69–70.
18. Abed V, Kapp S, Nichols M, Brunty N, Conley CEW, Jacobs CA, et al. ASES and UCLA Are Responsive Patient-Reported Outcome Measures After Rotator Cuff Repair: A Systematic Review and Meta-analysis. Am J Sports Med. 2024 Oct 17;52(12):3173–8. 

Reviewer 2 Report

Comments and Suggestions for Authors

Comments for Authors:

1.     Expand on the mechanisms by which US-PICT and ESWT are hypothesized to differ in effectiveness, particularly about calcification type.

2.     Highlight calcifying tendinopathy's global burden or prevalence to emphasize its significance beyond the studied population.

3.     Clarify how patient adherence to protocols, especially for ESWT, will be tracked and analyzed.

4.     Provide details on handling missing data due to dropouts or non-compliance beyond simply stating the use of SPSS.

5.     Explain the rationale for excluding calcifications smaller than 5mm or using only the Bianchi Martinolli classification, if relevant.

6.     Enhance the study flowchart (Scheme 1) to include sample size at each stage (e.g., recruitment, randomization, and follow-ups).

7.     Include a description of how adverse events will be monitored and reported.

8.     While the statistical methods are appropriate, details on handling missing data should be included to ensure transparency.

9.     Expand on how the findings might contribute to clinical practice, particularly in the absence of existing protocols for managing chronic CT.

10.  Review minor grammatical inconsistencies and rephrase long sentences for improved readability.

11.  Ensure consistent use of abbreviations and terminology throughout the manuscript (e.g., VAS vs. Visual Analogue Scale).

12.  Cite studies supporting the validity of tools like PGI-I and CGI-C in chronic pain conditions, even if tangentially related.

13.  Combined points about handling missing data into one cohesive comment.

14.  Grouped related comments under appropriate sections (e.g., methods, discussion).

15.  Streamlined overlapping points about protocol adherence and adverse event monitoring.

Author Response

Dear Reviewer:
Thank you very much for taking the time to review this manuscript. We also thank you very much for so kindly acknowledging the efforts we have made to standardize our intervention.
Please find the detailed responses below and the corresponding corrections highlighted in red in track changes in the re-submitted files.

Comments 1: Expand on the mechanisms by which US-PICT and ESWT are hypothesized to differ in effectiveness, particularly about calcification type. 
Response 1:  Thank you very much for this point of view. We have reorganized the discussion by adding new referenced information that we hope you will like. Among these changes we highlight the information about the mechanism of action of the ESWT between lines 381 -382 and the US – PICT between 400 - 402. In addition, it has been expressed in lines 429 - 430 “the impossibility of finding comparisons between these techniques based on the stage of the injury.”
De la Corte-Rodríguez H, Román-Belmonte JM, Rodríguez-Damiani BA, Vázquez-Sasot A, Rodríguez-Merchán EC. Extracorporeal Shock Wave Therapy for the Treatment of Musculoskeletal Pain: A Narrative Review. Healthc. 2023;11(21):2830. 
Albano D, Gambino A, Messina C, Chianca V, Gitto S, Faenza S, et al. Ultrasound-Guided Percutaneous Irrigation of Rotator Cuff Calcific Tendinopathy (US-PICT): Patient Experience. Biomed Res Int. 2020 Jun 10;2020:1–7.

Comments 2: Highlight calcifying tendinopathy's global burden or prevalence to emphasize its significance beyond the studied population.
Response 2: Dear reviewer.
Thank you very much for requesting this information. We consider that this information should be better highlighted by what we show: the prevalence of this pathology, expressed in lines 42 - 43, is “between 3 - 8% in the general population”.

Comments 3: Clarify how patient adherence to protocols, especially for ESWT, will be tracked and analyzed.
Response 3: Thank you for your interest in this point. The patient recruitment process has been explained in more detail in point 3.7 recruitment. The mechanisms used to refer patients to the different treatments and subsequent reviews will be carried out by the Reina Sofía Hospital mechanisms through DIRAYA. Dr. Javier Muñoz will be in charge of ensuring that the treatment and review times are met.

Comments 4: Provide details on handling missing data due to dropouts or non-compliance beyond simply stating the use of SPSS. 
Response 4:  Thank you very much for this feedback. It has been noted in lines 164 and 165 that the database created will not include patients with missing data.

Comments 5: Explain the rationale for excluding calcifications smaller than 5mm or using only the Bianchi Martinolli classification, if relevant. 
Response 5: Thank you for mentioning this aspect.
The limit size has been established as stated in the article by Sánchez Lite et al, in which they use this size. That is why, in order to resolve the doubts that have arisen regarding this data, this study has been referenced in the text on line 128.
Sánchez I, Toribio  B, Osorio S, Romera de Blas C, Andrés N. Treatment of calcific tendinopathy of the rotator cuff with ultrasound-guided puncture and aspiration. Radiología. 2023; 65(S2): 33–40.
Regarding the use of the Bianchi Martinolli classification, as it appears on lines 59-60, this classification “has been created in relation to the evolutionary course” of the CT

Comments 6:  Enhance the study flowchart (Scheme 1) to include sample size at each stage (e.g., recruitment, randomization, and follow-ups). 
Response 6 Thank you for this point of view. For us this flowchart is essential, so we want it to be as useful as possible. To do this we have added the (n=…) in each phase and we have established the exact moment of the blinding.

Comments 7: Include a description of how adverse events will be monitored and reported.
Response 7:  Dear reviewer, thank you for your comment on this section.
A new section has been created on 3.9 adverse effects between lines 221-226. This section explains the course of action in the event of the possible appearance of such effects.

Comments 8: While the statistical methods are appropriate, details on handling missing data should be included to ensure transparency.
Response 8: Thank you very much for this feedback. It has been noted in lines 164 and 165 that the database created will not include patients with missing data.

Comments 9: Expand on how the findings might contribute to clinical practice, particularly in the absence of existing protocols for managing chronic CT.
Response 9: Thank you for this feedback. To improve the information required, we have added a new reference in the discussion section between lines 419 - 421 that refers to "the reasonable need to test US-PICT and ESWT as treatment" prior to evaluating a surgical option.
Angileri HS, Gohal C, Comeau-Gauthier M, Owen MM, Shanmugaraj A, Terry MA, et al. Chronic calcific tendonitis of the rotator cuff: a systematic review and meta-analysis of randomized controlled trials com-paring operative and nonoperative interventions. J Shoulder Elbow Surg. 2023; 32(8):1746–60.

Comments 10: Review minor grammatical inconsistencies and rephrase long sentences for improved readability.
Response 10:  Thank you very much for this feedback. We consider the clarity of the text to be as good as possible in order to facilitate the expansion of the proposed protocol. We would like to know which grammatical inconsistencies should be revised.

Comments 11: Ensure consistent use of abbreviations and terminology throughout the manuscript (e.g., VAS vs. Visual Analogue Scale).
Response 11: Dear Reviewer, Thank you for pointing this out. We have reviewed all abbreviations so that they appear correctly in the article.

Comments 12: Cite studies supporting the validity of tools like PGI-I and CGI-C in chronic pain conditions, even if tangentially related.
Response 12: We appreciate that you pointed out this aspect, which can undoubtedly represent a limitation of the design in our research and as such we have added between lines 356 - 364 a more detailed explanation of why we consider it appropriate to use these scales. Currently, the PGI-I and CGI-C scales do not have validation for rotator cuff tendon pathologies. However, we believe that they can be very useful to us.
     We consider that it is important to take into account the results self-reported by the patients themselves in this type of pathology. 
      In Spain, some consensus documents have been prepared on the approach to chronic non-cancer pain in which they recommend the use of these scales. (17) Additionally, systematic reviews and meta-analyses can be found such as that by Varag Abed et al., in which they report that “Multiple patient-reported outcome measures have been used to evaluate shoulder function, but it is unknown which ones.” are the most effective.”(18)
17.Sánchez J, Tejedor A, Carrascal R. In Atención al Paciente con Dolor Crónico No Oncológico en AP. Sociedad Española de Médicos Generales y de Familia en AP. Spain, 2023; pp. 69–70. 
18. Abed V, Kapp S, Nichols M, Brunty N, Conley CEW, Jacobs CA, et al. ASES and UCLA Are Responsive Patient-Reported Outcome Measures After Rotator Cuff Repair: A Systematic Review and Meta-analysis. Am J Sports Med. 2024 Oct 17;52(12):3173–8. 

Comments 13: Combined points about handling missing data into one cohesive comment. 
Response 13:  Thank you very much for this feedback. It has been noted in lines 164 and 165 that the database created will not include patients with missing data.

Comments 14: Grouped related comments under appropriate sections (e.g., methods, discussion).
Response 14: We thank you for your observation and we have reviewed the content of the different sections of the magazine template to check if all the comments in the manuscript were in their corresponding section.

Comments 15: Streamlined overlapping points about protocol adherence and adverse event monitoring.
Response 15: Dear Reviewer. This information has been answered in comments 3 and 7.

Reviewer 3 Report

Comments and Suggestions for Authors

Dear Authors,

Thank you for the opportunity to review your study protocol titled "Calcifying Tendinopathy of the Rotator Cuff: Barbotage vs. Shock Waves - Controlled Clinical Trial Protocol (BOTCH)." This research is of significant clinical interest as it addresses an important and unresolved area in musculoskeletal rehabilitation. The manuscript demonstrates a well-structured clinical trial with clearly defined objectives, appropriate methodologies, and a comprehensive approach.

However, there are several areas where clarity and detail could be improved to align more closely with the SPIRIT 2013 guidelines. Below, I provide constructive feedback aimed at enhancing the rigor, transparency, and scientific quality of your protocol.

1. Title and Abstract

The title accurately conveys the study’s focus but could be strengthened by emphasizing the comparative nature of the interventions. Consider revising the title to something like: “Comparative Study of Ultrasound-Guided Barbotage vs. Shock Waves for Calcifying Tendinopathy: A Randomized Controlled Trial Protocol.”

The abstract provides a useful summary, though I recommend emphasizing the specific patient outcomes being measured, such as improvements in pain (VAS) and functionality (ROM). This will clarify the study’s endpoints for potential readers.

2. Introduction

The introduction effectively outlines the clinical relevance of calcifying tendinopathy and the rationale for comparing the two treatment modalities. However, the hypothesis could be stated more explicitly. For example:

"We hypothesize that ultrasound-guided barbotage will provide equal or superior pain relief and functional improvement compared to extracorporeal shockwave therapy in patients with calcifying tendinopathy of the rotator cuff."

This clearer framing would strengthen the scientific narrative. Additionally, consider expanding the background on how the treatments differ mechanistically to justify the study design further.

3. Objectives and Outcomes

The primary and secondary objectives are clearly stated. However, for the secondary objectives, it would be helpful to specify how improvements will be quantified, particularly for scales like PGI-I and CGI-C, which lack prior validation for this population. Consider justifying the use of these scales in the context of the study and referencing similar applications if available.

4. Study Design and Methodology

The study design is well-conceived, with a robust randomization process and appropriate sample size calculation. However, the manuscript would benefit from a more detailed description of patient flow through the study. Including a flowchart visualizing enrollment, randomization, interventions, and follow-up evaluations would greatly enhance reader understanding.

Additionally, the rationale for using a 1:1 block randomization should be explained more clearly. Was it chosen to balance patient characteristics between treatment arms or due to practical clinical considerations?

5. Interventions

The descriptions of the two interventions are thorough and technically precise. However, I recommend specifying how procedural consistency will be maintained across practitioners performing the interventions. Will there be specific training sessions or adherence monitoring? This will reassure readers that inter-operator variability has been minimized.

6. Data Collection and Management

The manuscript clearly outlines data management procedures, though details on data security and storage could be expanded. Consider specifying:

  • How data will be encrypted and securely stored.
  • Which members of the research team will have access to the data.
  • How missing data will be managed, particularly if patients withdraw prematurely.

7. Statistical Analysis

The proposed statistical analyses are appropriate and well-defined. However, assumptions behind the sample size calculation should be made explicit. For example, provide the expected effect size, standard deviation, and confidence intervals used in the calculation. This will help reviewers and future readers understand the study’s power and potential limitations.

8. Ethical Considerations and Dissemination Plans

The ethical framework is comprehensive, and the study has received institutional ethics approval. You could further strengthen this section by outlining your plans for public dissemination, including registering trial results in public databases and publishing findings in open-access journals.

Additionally, specifying how patients will be informed about study outcomes and how results will be shared with healthcare practitioners could increase the study's impact.

General Writing and Presentation

Overall, the manuscript is clearly written and well-organized. Consider simplifying some highly technical sections to make the protocol accessible to a broader readership. Minor grammatical adjustments would also improve flow.

Conclusion

In conclusion, this study protocol is well-designed and addresses a relevant clinical gap. My suggestions focus on enhancing clarity, justifying methodological choices, and providing additional context where necessary. I commend the authors for their thorough work and look forward to seeing the study's outcomes published.

Recommendation of Relevant Literature for Citation in the Manuscript

To strengthen the scientific foundation of your study protocol and enhance its contextual depth, I recommend citing the following article in specific sections of your manuscript. This article provide critical insights into ultrasound-guided interventions, tendon pathology assessments, and pain management, which is directly relevant to your research on barbotage and shockwave therapy for calcifying tendinopathy.

Article: "Effects of Orthopedic Manual Therapy on Pain Sensitization in Patients with Chronic Musculoskeletal Pain: An Umbrella Review with Meta-Meta-Analysis"

DOI: 10.1097/PHM.0000000000002239

Recommended Section: Discussion (Therapeutic Implications)

Reason for Citation:
This meta-analysis highlights the effectiveness of manual therapy techniques in reducing pain sensitization through central and peripheral mechanisms. Though your study focuses on barbotage and shockwave therapy, referencing this article will help contextualize your findings within the broader spectrum of non-surgical musculoskeletal treatments, emphasizing the multifactorial approach to pain management.

These citations will provide theoretical support and a broader scientific context for your study, enhancing its academic rigor and visibility within the field of musculoskeletal research.

Thank you for considering these recommendations. I believe they will help refine the manuscript and strengthen its contribution to the field of musculoskeletal research.

Kind regards,

Author Response

Dear Reviewer:
Thank you very much for taking the time to review this manuscript. We also thank you for your consideration of the clinical interest that our research may have and for your favourable comments on the design of our trial. Please find the detailed responses below and the corresponding corrections highlighted in red changes in the resubmitted files.

Comments 1: Title and Abstract
The title accurately conveys the study’s focus but could be strengthened by emphasizing the comparative nature of the interventions. Consider revising the title to something like: “Comparative Study of Ultrasound-Guided Barbotage vs. Shock Waves for Calcifying Tendinopathy: A Randomized Controlled Trial Protocol.”

The abstract provides a useful summary, though I recommend emphasizing the specific patient outcomes being measured, such as improvements in pain (VAS) and functionality (ROM). This will clarify the study’s endpoints for potential readers.

Response 1: Thanks for the opinion shown. We take this opinion with good pleasure. 

The team believes that he is correct in the comment expressed regarding the summary. We have clarified in line 23 of the article that the objectives focus on improving VAS and ROM.

Comments 2:     Introduction
The introduction effectively outlines the clinical relevance of calcifying tendinopathy and the rationale for comparing the two treatment modalities. However, the hypothesis could be stated more explicitly. For example:

"We hypothesize that ultrasound-guided barbotage will provide equal or superior pain relief and functional improvement compared to extracorporeal shockwave therapy in patients with calcifying tendinopathy of the rotator cuff."

This clearer framing would strengthen the scientific narrative. Additionally, consider expanding the background on how the treatments differ mechanistically to justify the study design further.

 Response 2: Regarding the second comment. We appreciate this assessment, which has been common to that of other critics.
For this reason, we have opted for a joint option and we have added new referenced information. 
Among these changes we highlight the information on the mechanism of action of the ESWT between lines 381 -382 and the US – PICT between 400 - 402.

De la Corte-Rodríguez H, Román-Belmonte JM, Rodríguez-Damiani BA, Vázquez-Sasot A, Rodríguez-Merchán EC. Extracorporeal Shock Wave Therapy for the Treatment of Musculoskeletal Pain: A Narrative Review. Healthc. 2023;11(21):2830. 
Albano D, Gambino A, Messina C, Chianca V, Gitto S, Faenza S, et al. Ultrasound-Guided Percutaneous Irrigation of Rotator Cuff Calcific Tendinopathy (US-PICT): Patient Experience. Biomed Res Int. 2020; 1–7.

Comments 3: Objectives and Outcomes
The primary and secondary objectives are clearly stated. However, for the secondary objectives, it would be helpful to specify how improvements will be quantified, particularly for scales like PGI-I and CGI-C, which lack prior validation for this population. Consider justifying the use of these scales in the context of the study and referencing similar applications if available.
Response 3: We appreciate that you pointed out this aspect, which can undoubtedly represent a limitation of the design in our research and as such we have added between lines 357 - 365 a more detailed explanation of why we consider it appropriate to use these scales. Currently, the PGI-I and CGI-C scales do not have validation for rotator cuff tendon pathologies. However, we believe that they can be very useful to us.
     We consider that it is important to take into account the results self-reported by the patients themselves in this type of pathology. 
      In Spain, some consensus documents have been prepared on the approach to chronic non-cancer pain in which they recommend the use of these scales. (17) Additionally, systematic reviews and meta-analyses can be found such as that by Varag Abed et al., in which they report that “Multiple patient-reported outcome measures have been used to evaluate shoulder function, but it is unknown which ones.” are the most effective.”(18)

17. Sánchez J, Tejedor A, Carrascal R. In Atención al Paciente con Dolor Crónico No Oncológico en AP. Sociedad Española de Médicos Generales y de Familia en AP. Spain, 2023; pp. 69–70.
18. Abed V, Kapp S, Nichols M, Brunty N, Conley CEW, Jacobs CA, et al. ASES and UCLA Are Responsive Patient-Reported Outcome Measures After Rotator Cuff Repair: A Systematic Review and Meta-analysis. Am J Sports Med. 2024;52(12):3173–8. 

Comments 4:     Study Design and Methodology
The study design is well-conceived, with a robust randomization process and appropriate sample size calculation. However, the manuscript would benefit from a more detailed description of patient flow through the study. Including a flowchart visualizing enrollment, randomization, interventions, and follow-up evaluations would greatly enhance reader understanding.
Additionally, the rationale for using a 1:1 block randomization should be explained more clearly. Was it chosen to balance patient characteristics between treatment arms or due to practical clinical considerations?
Response 4: The need for a flowchart that explains the protocol is a common point between you and our research team. 
That is why this flowchart is included on page 4. The flowchart has been modified by adding (n=...) in each phase and also specifying the moment of blinding of the data.
Regarding the allocation by blocks, we agree on the need to explain the reason for said allocation. 
For this we have added an explanatory paragraph between lines 201 -202 in which it is explained that "Randomization is established by randomization blocks, to avoid numerical inequality between the different groups and to obtain the sample size needed in each group"

Comments 5:     Interventions
The descriptions of the two interventions are thorough and technically precise. However, I recommend specifying how procedural consistency will be maintained across practitioners performing the interventions. Will there be specific training sessions or adherence monitoring? This will reassure readers that inter-operator variability has been minimized.

Response 5:  Thank you for your interest in this point. The patient recruitment process has been explained in more detail in point 3.7 recruitment. The mechanisms used to refer patients to the different treatments and subsequent reviews will be carried out by the Reina Sofía Hospital mechanisms through DIRAYA. Dr. Javier Muñoz will be in charge of ensuring that the treatment and review times are met.

Comments 6: Data Collection and Management
The manuscript clearly outlines data management procedures, though details on data security and storage could be expanded. Consider specifying:
How data will be encrypted and securely stored.
Which members of the research team will have access to the data.
How missing data will be managed, particularly if patients withdraw prematurely.

Response 6: Thank you very much for this appreciation.
Information regarding data encryption appears in section 4. Data Collection and Management. To this section we have added who has access to DIRAYA's data.
In relation to missing data, it has been noted in lines 164 and 165 that the database created will not include patients with missing data.
Comments 7:     Statistical Analysis
The proposed statistical analyses are appropriate and well-defined. However, assumptions behind the sample size calculation should be made explicit. For example, provide the expected effect size, standard deviation, and confidence intervals used in the calculation. This will help reviewers and future readers understand the study’s power and potential limitations.
 Response 7: We consider, as you, the importance of knowing the data to which you refer. 
That is why in section 3.6 Simple size, the referred data appears, among them: alpha risk, statistical power, type of contrast, the minimum difference to detect, the standard deviation and the percentage of losses.

Comments 8 Ethical Considerations and Dissemination Plans
The ethical framework is comprehensive, and the study has received institutional ethics approval. You could further strengthen this section by outlining your plans for public dissemination, including registering trial results in public databases and publishing findings in open-access journals.
Additionally, specifying how patients will be informed about study outcomes and how results will be shared with healthcare practitioners could increase the study's impact.
Response 8: The purpose of every article is to achieve the greatest possible scientific dissemination. That is why this team agrees with the above assessment. 
To do this, we have created a new section “Data Availability   Statement” between lines 472 - 476, which states the following: " The database obtained for the analysis of this study will be made available to those researchers who request it. 
When the data is analyzed, it will be published in full detail in open access, except for those that affect the confidentiality of the participants. On this database, any type of personal information that allows patients to be identified will be eliminated."

Recommendation of Relevant Literature for Citation in the Manuscript
This team thanks you for the assessments made. We agree with them. 
Regarding bibliographic recommendations, since this study belongs to the doctoral thesis of our main author, which focuses on second-line treatments for rotator cuff pathologies, they will be considered for reference in future publications.

Reviewer 4 Report

Comments and Suggestions for Authors

Dear authors,

Congratulations on your clinical trial proposal. I consider that your proposal is of interest and may contribute to the development of rotator cuff treatments. I hope that in addition to this communication you can develop the complete clinical trial.

The following comments are recommendations that could help improve your article:

- In the title and in the first appearance in the text it would be convenient to indicate the meaning of the acronym BOTCH.

- The content of the introduction is adequate and explains well the predisposing factors and the evaluation with diagnostic imaging means of calcific tendinopathies. However, it could be advisable to include more information on the current conceptualization of tendinopathies. References to authors such as Cook and Alfredson are missing.

- Following the same line, both in the introduction and in the discussion it would be convenient to relate the proposed treatments with the associated physiotherapy intervention. 

- It might be advisable to include content on the types of therapeutic exercise to be included as an associated treatment. It would also be appropriate to cite research that proposes physiotherapy guidelines beyond conventional approaches (Góngora-Rodríguez J, Rosety-Rodríguez MÁ, Rodríguez-Almagro D, Martín-Valero R, Góngora-Rodríguez P, Rodríguez-Huguet M. Structural and Functional Changes in Supraspinatus Tendinopathy through Percutaneous Electrolysis, Percutaneous Peripheral Nerve Stimulation and Eccentric Exercise Combined Therapy: A Single-Blinded Randomized Clinical Trial. Biomedicines. 2024 Mar 30;12(4):771. doi: 10.3390/biomedicines12040771.)

The content you have shown as diagram 1 is remarkable, because it contributes to a very visual understanding of the study protocol. Congratulations.

- Finally, with respect to the assessment instruments included in the methodology, specific functional assessment scales that relate pain to the functional capacity of the upper limb could also be included, such as the DASH, WOSI or SPADI scales, which have a validated version in Spanish.

Author Response

Dear Reviewer:
Thank you very much for taking the time to review this manuscript. We also thank you for your consideration of the clinical interest that our research may have. Please find the detailed responses below and the corresponding corrections highlighted in red changes in the resubmitted files.

Comments 1: In the title and in the first appearance in the text it would be convenient to indicate the meaning of the acronym BOTCH.

Response 1: We agree with this assessment and have therefore added a paragraph on lines 80 to 83 explaining that "this protocol, whose acronym comes from the Spanish words “barbotaje, onda de choque, tendinopatía, calcio y hombro” (BOTCH), will focus its research on the comparison of second-line CT treatments such as ultrasound-guided barbotage (US-PICT) and shock waves (ESWT).”

Comments 2:     The content of the introduction is adequate and explains well the predisposing factors and the evaluation with diagnostic imaging means of calcific tendinopathies. However, it could be advisable to include more information on the current conceptualization of tendinopathies. References to authors such as Cook and Alfredson are missing.

 Response 2: Thank you very much for this assessment.

It is essential for us to search for possible theories on the etiology of this type of pathology, so we have added a reference on lines 48 - 49 which refers to the continuum theory proposed by Cook. 

Cook JL, Purdam CR. Is tendon pathology a continuum? A pathology model to explain the clinical presentation of load-induced tendinopathy. Br J Sports Med. 2009;43(6):409–16.

Comments 3: Following the same line, both in the introduction and in the discussion, it would be convenient to relate the proposed treatments with the associated physiotherapy intervention. 
Comments 4:   It might be advisable to include content on the types of therapeutic exercise to be included as associated treatment. It would also be appropriate to cite research that proposes physiotherapy guidelines beyond conventional approaches (Góngora-Rodríguez J, Rosety-Rodríguez MÁ, Rodríguez-Almagro D, Martín-Valero R, Góngora-Rodríguez P, Rodríguez-Huguet M. Structural and Functional Changes in Supraspinatus Tendinopathy through Percutaneous Electrolysis, Percutaneous Peripheral Nerve Stimulation and Eccentric Exercise Combined Therapy: A Single-Blinded Randomized Clinical Trial. Biomedicines. 2024 Mar 30;12(4):771. doi: 10.3390/biomedicines12040771.)  
Response 3 and 4: Dear reviewer, regarding comments 3 and 4.

The working team agrees on the importance of physiotherapeutic intervention therapies in this type of pathology. However, it is true that in this protocol the intervention carried out by the physiotherapist is the realization of ESWT.
No other types of additional therapies have been added (ultrasound, thermotherapy, kinesiotherapy, etc.) in order to know in detail, the results of these two techniques.
Having said this, in the publication of the final results, emphasis will be placed on the importance of adding exercises and other additional interventions. We take note of the article proposed for future publication.

Comments 5:   The content you have shown as diagram 1 is remarkable, because it contributes to a very visual understanding of the study protocol. Congratulations.  

Response 5: We appreciate this appreciation, since we believe in the necessity of this type of flowcharts for the understanding of the study.

Comments 6: Finally, with respect to the assessment instruments included in the methodology, specific functional assessment scales that relate pain to the functional capacity of the upper limb could also be included, such as the DASH, WOSI or SPADI scales, which have a validated version in Spanish.
Response 6: Thank you very much for this assessment.
We know the importance of using validated scales, and the proposed ones are the most used. Initially, their use was proposed, but due to the limited time available for consultation, the team decided to use only the EVA, the ROM and the Latineen as the main measures, which are also validated for chronic pain. We agree with you that having proposed the use of these scales would have improved the quality of the study in relation to shoulder pathologies.
However, this study belongs to the doctoral program of our main author, who will use the Spanish version of these validated scales in future studies related to rotator cuff pathology.